# The role of competition versus cooperation in microbial community coalescence

**Pablo Lechón-Alonso**[1,2]*, **Tom Clegg**[2], **Jacob Cook**[2,3], **Thomas P. Smith**[2], **Samraat Pawar**[2]

**1** Department of Ecology & Evolution, University of Chicago, Chicago, Illinois, United States of America, **2** Department of Life Sciences, Imperial College London, Ascot, United Kingdom, **3** Centre for Integrative Systems Biology and Bioinformatics, Imperial College London, London, United Kingdom

* plechon@uchicago.edu

**Data Availability Statement:** All the code used in our simulations, as well as that for reproducing the figures can be found in the GitHub repository: https://github.com/pablolich/coalescence_paper_analysis.

## Abstract

New microbial communities often arise through the mixing of two or more separately assembled parent communities, a phenomenon that has been termed "community coalescence". Understanding how the interaction structures of complex parent communities determine the outcomes of coalescence events is an important challenge. While recent work has begun to elucidate the role of competition in coalescence, that of cooperation, a key interaction type commonly seen in microbial communities, is still largely unknown. Here, using a general consumer-resource model, we study the combined effects of competitive and cooperative interactions on the outcomes of coalescence events. To do so, we simulate coalescence events between pairs of communities with different degrees of competition for shared carbon resources and cooperation through cross-feeding on leaked metabolic by-products (facilitation). We also study how structural and functional properties of post-coalescence communities evolve when they are subjected to repeated coalescence events. We find that in coalescence events, the less competitive and more cooperative parent communities contribute a higher proportion of species to the new community because of their superior ability to deplete resources and resist invasions. Consequently, when a community is subjected to repeated coalescence events, it gradually evolves towards being less competitive and more cooperative, as well as more speciose, robust and efficient in resource use. Encounters between microbial communities are becoming increasingly frequent as a result of anthropogenic environmental change, and there is great interest in how the coalescence of microbial communities affects environmental and human health. Our study provides new insights into the mechanisms behind microbial community coalescence, and a framework to predict outcomes based on the interaction structures of parent communities.

## Author summary

In nature, new microbial communities often arise from the fusion of whole, previously separate communities (community coalescence). Despite the crucial role that interactions among microbes play in the dynamics of complex communities, our ability to predict how

**Funding:** This work was supported by the Leverhulme Research Fellowship (RF-2020-653\2) and NERC grant NE/S000348/1.

**Competing interests:** The authors have declared that no competing interests exist.

these affect the outcomes of coalescence events remains limited. Here, using a general mathematical model, we study how the structure of species interactions confers an advantage upon a microbial community when it encounters another, and how communities evolve after undergoing repeated coalescence events. We find that competitive interactions between species preclude their survival upon a coalescence event, while cooperative interactions are advantageous for post-coalescence survival. Furthermore, after a community is exposed to many coalescence events, the remaining species become less competitive and more cooperative. Ultimately, this drives the community evolution, yielding post-coalescence communities that are more species-rich, productive, and resistant to invasions. There are many potential environmental and health implications of microbial community coalescence, which will benefit from the theoretical insights that we offer here about the fundamental mechanisms underlying this phenomenon.

## Introduction

Microbial communities are widespread throughout our planet [1], from the human gut to the deep ocean, and play a critical role in natural processes ranging from animal development and host health [2, 3] to biogeochemical cycles [4]. These communities are very complex, typically harboring hundreds of species [5], making them hard to characterize. Over the last decade, the advent of cost-effective high-throughput DNA sequencing techniques has allowed high-resolution mapping of these communities, opening a niche for theoreticians and experimentalists to collaboratively decipher their complexity and assembly [6–12].

Entire microbial communities are often displaced over space and come into contact with each other due to physical (e.g., dispersal by wind or water) and biological (e.g., animal-animal or animal-plant interactions, and leaves falling to the ground) factors [13–16]. The process by which two or more communities that were previously separated join and reassemble into a new community has been termed community coalescence [17]. Although microbial community coalescence is likely to be common, the effects of both intrinsic and extrinsic factors on the outcomes of such events remains poorly understood [18]. Among extrinsic factors, resource availability, immigration rate of new species, and environmental conditions (especially, pH, temperature, and humidity) are likely to be crucial [19–21]. Among intrinsic factors, the role of functional and taxonomic composition and the inter-species interaction structures of parent communities are expected to be particularly important [19, 20]. We focus on the role of species interactions on community coalescence in this study.

Early mathematical models suggested that in encounters between animal and plants communities, species in one community are more likely to drive those in the other extinct (community dominance) [22, 23]. This was explained as being the result of the fact that communities are a non-random collection of species assembled through a shared history of competitive exclusion, and therefore act as coordinated entities. Recent theoretical work [24] has more rigorously established this for microbial community coalescence events, showing that the dominant community will be the one capable of more efficiently depleting all resources simultaneously. Overall, these findings suggest that communities arising from competitive species sorting exhibit sufficient "cohesion" to prevent invasions by members of other communities [25, 26].

However, empirical support for the role of competition alone in coalescence outcomes is circumstantial, and the role of cooperation, which is commonly observed in microbial communities, is yet to be addressed theoretically. For example, during coalescence in

methanogenic communities, "cohesive" units of taxa from the community with the most efficient resource use are co-selected [21]; and in aerobic bacterial communities, the invasion success of a given taxon is determined by its community members as a result of collective consumer-resource interactions and metabolic feedbacks between microbial growth and the environment [27]. These microbial communities exhibit cooperation through a typically dense cross-feeding network, where leaked metabolic by-products of one species are shared as public goods across the entire community [28–30]. Nonetheless, neither [21] nor [27] addressed the role of competition and cooperation in determining the outcome of coalescence events in microbial communities, which have been suggested as key factors that may influence it [20, 31, 32].

Here, we first use a general consumer-resource model that includes cross-feeding to assemble complex microbial communities having different degrees of competition and cooperation. Thereafter, we focus on determining the relative importance of the two types of interactions on outcomes of coalescence events between the assembled communities. Finally, we quantify the subsequent evolution of the structural and functional properties of coalesced communities.

## Methods

### Mathematical model

Our mathematical model for the microbial community dynamics is based on the work of Marsland et al. [6] (see Section 1 in S1 Text, Table 1, and Fig 1):

$$\frac{dN_\alpha}{dt} = g_\alpha N_\alpha \left( (1 - l_\alpha) \sum_j c_{\alpha j} R_j - z_\alpha \right),$$

$$\frac{dR_j}{dt} = \kappa_j + \tau^{-1} R_j - \sum_\alpha N_\alpha c_{\alpha j} R_j + \sum_{\alpha k} N_\alpha l_\alpha D_{\alpha k j} c_{\alpha k} R_k. \tag{1}$$

Here, $N_\alpha$ ($\alpha = 1, \ldots, s$) and $R_j$ ($j = 1, \ldots, m$) are the biomass abundance (measured in arbitrary units of biomass) of the $\alpha^{th}$ microbial (e.g., bacterial) species and the concentration

**Table 1. Table of parameters used in our model.** The units $E_a$ and $B_a$ represent arbitrary energy and biomass units, respectively.

| Parameter | Description | Value | Units |
|:---:|:---|:---:|:---:|
| $g_\alpha$ | Biomass synthesised per unit energy harvested by species $\alpha$ | 1 | $B_a\, E_a^{-1}\, mol^{-1}$ |
| $c_{\alpha j}$ | Energy uptake rate of metabolite $j$ by species $\alpha$ | {1, 0} | $E_a\, s^{-1}\, B_a^{-1}$ |
| $D_{kj}$ | Fraction of metabolite $k$ that's leaked in the form of $j$ | [0, 1] | None |
| $\kappa_j$ | Supply rate of metabolite $j$ | 2 | $mol\, s^{-1}$ |
| $\tau$ | Time scale of resource dilution | 0.25 | s |
| $l_\alpha$ | Fraction of energy intake leaked as metabolic byproducts | [0, 1) | None |
| $\chi_0$ | Average cost per metabolic pathway | 0.1 | $E_a$ |
| $\epsilon_\alpha$ | Random fluctuation in species $\alpha$'s cost | N(0, 0.1) | None |
| $s$ | Number of species in pre-assembly communities | 60 | - |
| $m$ | Number of metabolites | 60 | - |
| $k_c$ | Competition factor | [0,1) | - |
| $k_f$ | Facilitation factor | [0,1] | - |
| $K_c$ | Inter-guild competition factor | (0.1, 0.9) | - |
| $K_f$ | Inter-guild facilitation factor | (0.1, 0.9) | - |

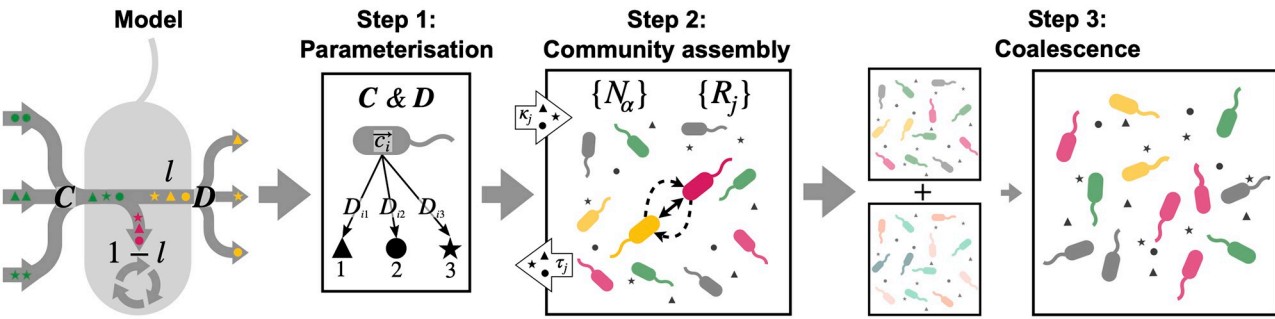

**Fig 1. Overview of the coalescence modelling methodology. Step 1**. The matrix of resource preferences ($C$) and the metabolic matrix ($D$) are sampled for each community. Black polygons are different resource types. **Step 2**. Dynamics of the system are allowed to play out (Eq 1) until a locally stable equilibrium is reached. Species composition and abundance, along with community-level competition $\mathcal{C}$ (solid bidirectional arrows, Eq 3), and facilitation $\mathcal{F}$ (dashed unidirectional arrows, Eq 4) are measured in assembled communities. **Step 3**. Two of the assembled parent communities are mixed, and the resulting community integrated to steady state. For the random and recursive coalescence procedures, the contribution of each parent community to the final mix is analyzed ($S_{1,2}$, Eq 7) as a function of their interaction structures ($\mathcal{C}_{1,2}$ and $\mathcal{F}_{1,2}$) before they coalesced. In the case of the serial coalescence procedure, the properties of the resident community $\mathcal{R}$ are tracked after each coalescence exposure.

(measured in moles) of the $j^{th}$ resource (e.g., carbon substrate). The growth of species $\alpha$ is determined by the energy it harvests minus its maintenance cost (two terms in the brackets). Energy is obtained through resource consumption, which depends on the resource concentration in the environment, $R_j$, and on the energy uptake rate per unit of biomass of species $\alpha$, $c_{\alpha j}$ (here assumed to be binary; either 1 or 0). The leakage term $l_\alpha$ determines the proportion of this uptake that species $\alpha$ releases back into the environment as metabolic by-products, with the remainder $(1 - l_\alpha)$ being allocated to growth. The energy surplus that remains after subtracting a maintenance cost ($z_\alpha$) is transformed into biomass with a proportionality constant of $g_\alpha$, the value of which does not affect the results presented here.

The change in the concentration of resources in the environment (second line in Eq 1) is determined by four terms. The first and second terms represent the external supply and dilution of resource $j$, which give the rates at which the $j^{th}$ resource enters and leaves the system. The third term is the uptake of the $j^{th}$ resource from the environment, summed across all $s$ consumers, and the fourth term represents resources entering the environmental pool via leakage of metabolic by-products. By-product leakage of species $\alpha$ is determined by the metabolic matrix $D_\alpha$ (or the "stoichiometric" matrix; [6]), with $D_{\alpha jk}$ representing the leaked proportion of resource $j$ that is transformed into resource $k$ by species $\alpha$. Energy conservation dictates that $D_\alpha$ is a row stochastic matrix, meaning that its rows sum to 1. Note that in this model, we allow species to leak metabolites that they also consume (we address this assumption further in the Discussion).

We define the consumer's maintenance cost to be:

$$z_\alpha = \chi_0 (1 + \epsilon_\alpha)(1 - l_\alpha) \sum_j c_{\alpha j}, \tag{2}$$

where $\chi_0$ is the average energetic cost of being able to consume a given resource, the summation represents the total number of resources that species $\alpha$ can process, and $\epsilon_\alpha$ is a small random fluctuation that introduces variation in the cost for species that have identical preferences. Eq 2 ensures that neither generalists nor specialists are systematically favored during community assembly (by imposing a greater cost on species that consume a wider range of resources) and that all species can deplete resources to similar concentrations independently

of their leakage level (see Section 1 in S1 Text for rationale, Fig A in S1 Text for results under different cost functions; and Discussion).

The above model entails the following assumptions: (i) all resources contain the same amount of energy (taken to be 1 for simplicity), (ii) a linear, non-saturating consumer functional response, (iii) binary consumer preferences (uptake rates), and (iv) an environment where all resources are externally supplied in equal amounts. We address the implications of these assumptions in the Discussion.

## Competition and facilitation metrics

In our model (Eq 1), competition for resources exists because all pairs of consumer species generally share some resource preferences (their metabolic preferences vectors are not orthogonal). We quantify the pairwise competition between a species pair $(\alpha, \beta)$ by counting the resource preferences they share through the scalar product of their preference vectors, $\boldsymbol{c}_\alpha \cdot \boldsymbol{c}_\beta$. Therefore, community-level competition (denoted as $\mathcal{C}$) can be calculated by taking the average of the competition matrix, which encodes the competitive strength between all species pairs, that is

$$\mathcal{C} = \langle CC^T \rangle, \tag{3}$$

where $C$ is the $s \times m$ matrix of metabolic preferences of all the species in the community.

On the other hand, facilitation occurs when a species leaks metabolic by-products that are used by another species. We measure pairwise facilitation from species $\alpha \rightarrow \beta$ by calculating the fraction of secreted resources from species $\alpha$ that are consumed by species $\beta$ per unit of resource abundance, $l_\alpha \boldsymbol{c}_\alpha^T D_\alpha \boldsymbol{c}_\beta$. Similar to competition, we compute community-level cooperation (denoted as $\mathcal{F}$), by taking the average of the facilitation matrix, which encodes the cooperative strength between all species pairs, that is

$$\mathcal{F} = \langle \sum_\alpha \mathcal{D}(\boldsymbol{l}) C D_\alpha C^T \rangle, \tag{4}$$

where where $\mathcal{D}(\boldsymbol{l})$ is a diagonal matrix with the leakage vector of each species in the community in its diagonal.

Henceforth, we refer to the quantity $\mathcal{C} - \mathcal{F}$ as "net competition", which we later show is related to the "cohesion" defined in previous work [24].

## Simulations

In Fig 1 we present an overview of our simulations, which we now describe. For the parameter values used, see Table 1.

**Step 1: Parameterization.** We first set the parameters of the initial communities (before assembly) such that they span interactions across the spectrum of net competition ($\mathcal{C} - \mathcal{F}$). For each parent community, we modulate the structure of the $C$ and $D$ matrices (consisting of the resource preferences $c_{\alpha j}$'s and secretion proportions $D_{jk}$'s, respectively) by developing constrained random sampling procedures that guarantee specific levels of competition and facilitation at the community's steady state. In particular, competition is modulated through the competition factor $k_c$, which is imposed during the sampling of $C$. When $k_c = 0$ the elements of $C$ are sampled uniformly at random, and as $k_c \rightarrow 1$ those resources that have been frequently sampled by other species are more likely to be assigned as one of the preferences of the current one (see Section 2 in S1 Text), thus generating hubs of highly demanded resources, which accentuate resource competition in the community. Net competition can alternatively be reduced through the effect of facilitation. We modulate facilitation using the facilitation factor

$k_f$, during the sampling of $D$. When $k_f \rightarrow 0$ the metabolic matrix has no structure, that is, all resources are released at equiprobable fractions. As $k_f \rightarrow 1$ the structure of $D$ becomes fully determined by the resource demands of the community so that more demanded resources are released at higher fractions (see Section 2 in S1 Text). This alleviates resource competition by creating a flow of resources from least to most demanded. In addition, we also add structure to $C$ and $D$ to emulate the existence of distinct resource classes and consumer guilds (see Section 4 in S1 Text). In these communities, competition and facilitation between guilds are modulated through the constants $K_c$ (inter-guild competition factor), and $K_f$ (inter-guild facilitation factor) (see Section 2 in S1 Text). Note that we parameterize the initial communities by assuming (i) a shared core metabolism encoded in $D$, and (ii) a common leakage fraction $l$ for all species (the implications of which we address in the Discussion), but we relax these assumptions in our coalescence simulations (Methods; Step 3).

**Step 2: Assembly of parent communities.** After parameterization, we numerically integrate Eq 1 until steady state (a putative equilibrium) is reached. We perform 100 such assembly simulations with random sets of consumers for each combination of competition and facilitation factors (i.e., $k_c = k_f \in [0, 0.5, 0.9]$), repeating this for three values of leakage ($l \in [0.1, 0.5, 0.9]$). We compare species composition, abundances, and interaction structure of communities before and after assembly. To compare species composition, we calculate the difference, before and after assembly, in the proportion of species with $m_r$ metabolic preferences (that is, a species with $m_r$ ones and $m - m_r$ zeros in its vector of metabolic preferences $\vec{c}$, which we now denote as an m-preferences consumer) as

$$\Delta m_r = \frac{1}{p(m_r)} \left( \frac{T_{m_r}}{r} - \frac{T_{m_r}^0}{r^0} \right). \tag{5}$$

Here, $p(m_r)$ is the probability that a species has $m_r$ metabolic preferences (which is exponentially distributed; Section 2 in S1 Text), the 0 denotes before assembly, $r$ is species richness, and $T_{m_r}$ is the number of species in the community with $m_r$ metabolic preferences. Thus, when $\Delta m_r > 0$ the proportion of species with $m_r$ metabolic preferences increases after assembly and vice versa. To analyze species abundances, we track the abundance fraction of consumers in each group of m-preferences species, calculated as the total abundance of species in the group, divided by the total community biomass, that is

$$w_{m_r} = \frac{\sum_{i \in m_r} n_i}{\sum_i n_i} \tag{6}$$

Finally, we address the interaction structure after assembly by quantifying the levels of competition ($\mathcal{C}$) and facilitation ($\mathcal{F}$) in the assembled communities (Fig 2A).

**Step 3: Coalescence.** The coalescence between a pair of assembled parent communities is simulated by setting all resources to their initial concentrations and numerically integrating the new combined system to steady state. To disentangle the effects of competition versus cooperation and study the effect of repeated coalescence events, we simulate three coalescence scenarios: random, recursive, and serial, as follows (further details in Section 3 in S1 Text).

**Random coalescence**. To address the effects of competition alone in the outcome of coalescence events, here we coalesce pairs of randomly sampled parent communities having the same leakage value $l$ ($2 \cdot 10^4$ pairs for each leakage level, Fig 3C). That is, we fix the leakage level to ensure that the communities have, on average, similar cooperation levels, but leave $k_c$ free to vary such that they span a broad range of competition levels.

**Recursive coalescence**. To study the effects of cooperation in particular on community coalescence, we repeatedly coalesce a given pair of communities $A$ and $B$, slightly increasing the

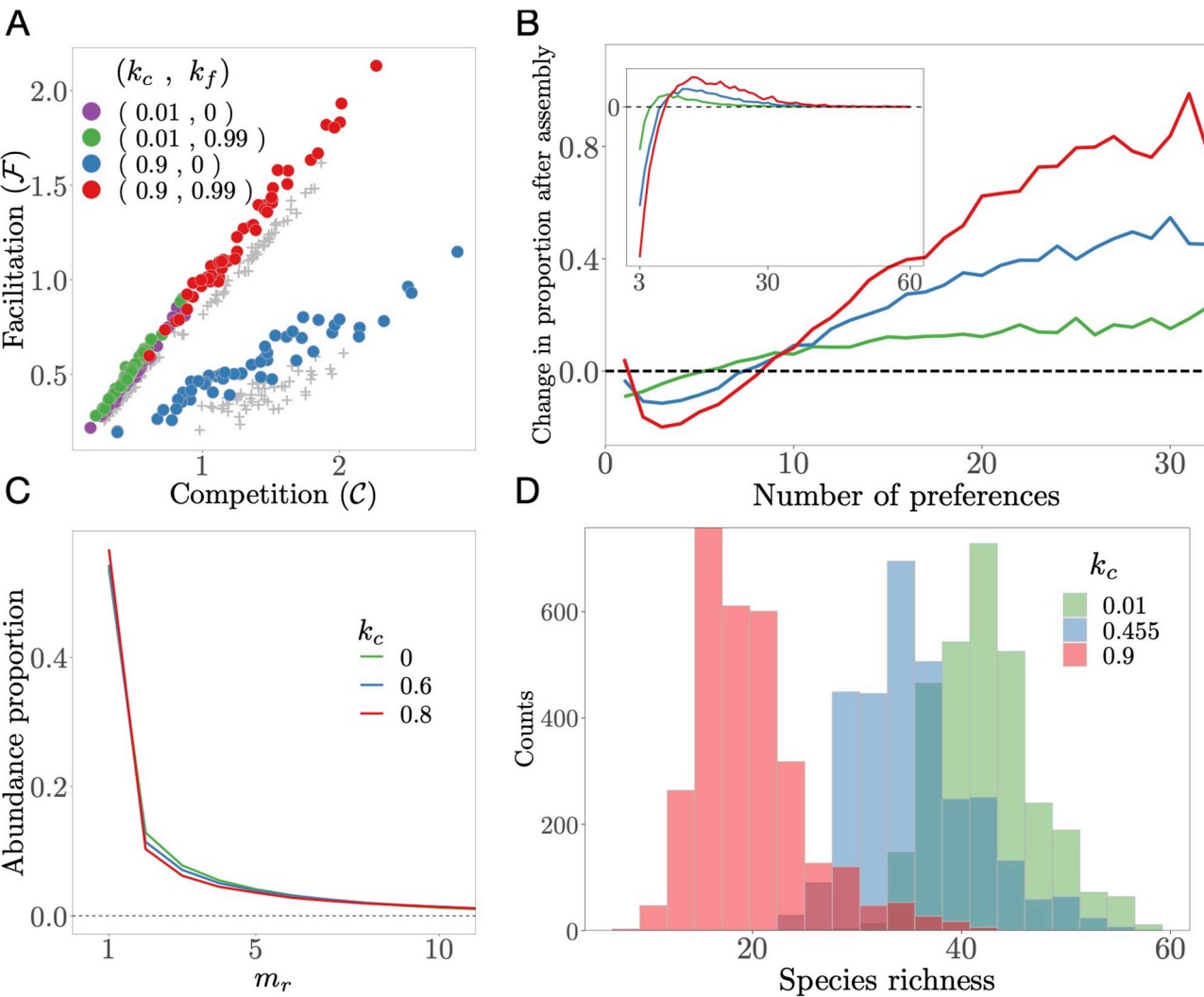

**Fig 2. Features of assembled parent communities. A**: Facilitation versus competition level in starting (grey dots) and assembled (coloured dots) communities for leakage $l = 0.9$ and different combinations of competition ($k_c$) and facilitation ($k_f$) factors. Communities assembled for each pair of [$k_c$, $k_f$] values have the same colour. The assembled communities are displaced towards more cooperative states at the end of the assembly than at the start. **B**: Change in proportion of species m-preferences consumers before and after assembly ($\Delta m_r$, Eq 5, y axis) as a function of the number of metabolic preferences (x axis), for three different values of $k_c$ (legend in panel C), calculated after each simulation, and averaged across simulations. Greater positive values of $\Delta m_r$ for more generalist species indicates that they are less prone to extinction during assembly. Values for $m_r > 30$ had too much uncertainty due to low sampling and therefore have been removed for clarity. **Inset**: $\Delta m_r$ is weighted by the abundance fraction at equilibrium of each m-preferences consumers group, $w_{m_r}$ (Eq 6). **C**: Abundance fraction of the m-preferences species groups for different values of $k_c$. **D**: Distributions of species richness values of parent communities assembled under different $k_c$ values. Increasing competitiveness tends to decrease species richness.

leakage of the latter in each iteration (Fig 4A). This allows us to modify the strength of cooperative interactions in the community, because facilitation is proportional to $l$ (Eq 4) while keeping competition levels constant because competition is independent of $l$ (Eq 3), and the remaining parameters are kept fixed.

**Serial coalescence**. To understand how the functional and structural properties of a microbial community evolve under successive coalescence events, we simulate the following scenario. A local ("resident") community $\mathcal{R}$ harbouring species with leakage $l_{\mathcal{R}}$, and metabolism $D_{\mathcal{R}}$ is successively invaded by many other randomly sampled communities ("invaders"), $\mathcal{I}$ with species of leakage $l_{\mathcal{I}}$ and metabolism $D_{\mathcal{I}}$ (Fig 5A).

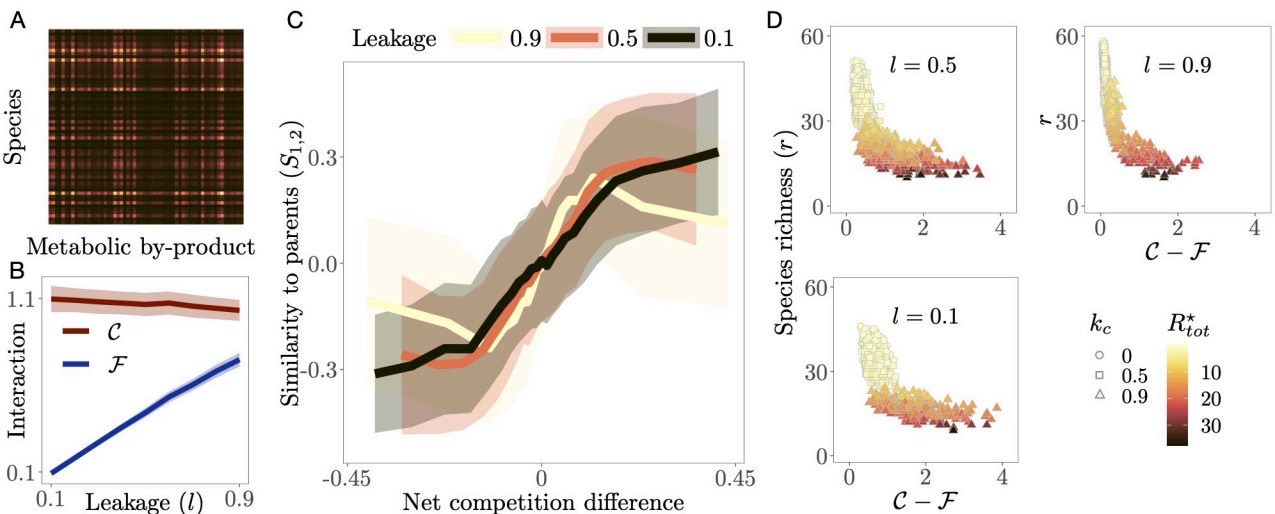

**Fig 3. Community coalescence between pairs of randomly picked communities with same leakage. A**: Example of the secretion matrix with elements $(CD)_{\alpha k}$ representing the total leakage of resource $k$ by species $\alpha$. **B**: Community-level competition $\mathcal{C}$ (dark red) and facilitation $\mathcal{F}$ (blue) averaged across simulations for each leakage value. Since competition does not depend on the leakage, it remains consistently high throughout. Facilitation, on the other hand, increases linearly with leakage. **C**: Parent community dominance ($S_{1,2}$) as function of net competition difference $(\mathcal{C}_1 - \mathcal{F}_1) - (\mathcal{C}_2 - \mathcal{F}_2)$ (solid lines ± 1 standard deviation (shaded)), binned (20 bins) over communities with similar x axis values, for three community-wide leakage levels. The post-coalescence community is more similar to its less net competitive parent. **D**: Species richness ($r$) as a function of net competition in parent communities, coloured by total resource concentration at steady state ($R_{tot}^\star$). The observed negative correlation for all values of leakage shows that communities with lower net competition tend to be more speciose and also better at depleting resources (brighter coloured values, corresponding to lower levels of $R_{tot}^\star$ are scattered towards the top left of the plots).

To determine which parent community is more successful after each random and recursive coalescence simulations, we measure the similarity between the post-coalescence community and each of the two parents (indexed by 1 and 2) as:

$$S_{1,2} = \boldsymbol{p}_f \cdot \left( \frac{\boldsymbol{p}_2}{r_2} - \frac{\boldsymbol{p}_1}{r_1} \right), \tag{7}$$

where $\boldsymbol{p}_f$, $\boldsymbol{p}_1$, and $\boldsymbol{p}_2$ are $(s_1 + s_2)$–dimensional binary vectors of species presence-absence in the post-coalescent, and parent communities 1 and 2, respectively, with $r_1$ and $r_2$ the species richness values of the parent communities 1 and 2, respectively (calculated as $r_i = \sum p_i$). If $S_{1,2} = -1$, the coalesced community is identical to parent community 1, and if $S_{1,2} = 1$, it is identical to parent community 2. This measure is independent of the species richness. Thus we can mix communities with different species richness while avoiding a bias in similarity towards the richer one. We then analyze how this dominance measure depends on the interaction structure of the parent communities ($\mathcal{C}_{1,2}$ and $\mathcal{F}_{1,2}$; Eqs 3 and 4). After each coalescence event in the serial coalescence procedure, we measure competition and facilitation levels of the resident community, along with the average species maintenance cost, average resource abundance at equilibrium, species richness, and number of successful invasions, during the entire sequence of serial coalescence events.

For all assembled parent as well as coalesced communities we confirmed that the steady state was a locally asymptotically stable equilibrium (Section 1 in S1 Text).

## Results

### Assembly of parent communities

In Fig 2 we show the key features of assembled communities. Fig 2A shows that as expected from Eqs 3 and 4, the levels of community-wide competition and facilitation are positively

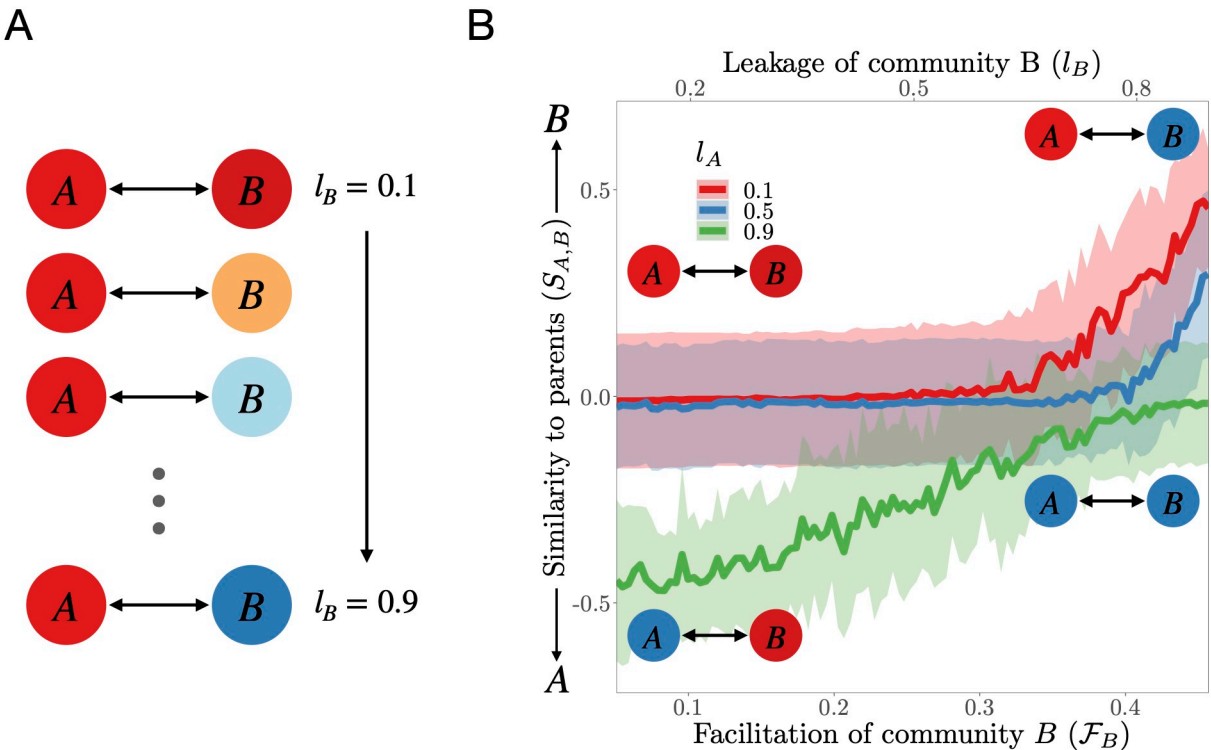

**Fig 4. Recursive coalescence between microbial communities. A**: Sketch of the simulation set up. The same pair of communities $(A, B)$ is recursively coalesced, with the leakage of $B$ gradually increasing after each coalescence event for three levels of leakage of A, and 25 replicates per $l_A$ value. **B**: Parent community dominance after coalescence between communities $A$ and $B$, as a function of facilitation level of community $B$, $\mathcal{F}_B$ (bottom x axis), and leakage of community $B$, $l_B$ (top x axis). Each curve corresponds to a different value of $l_A$. Shaded regions are $\pm\sigma$. Dominance of parent community $B$ after coalescence increases with $l_B$, implying that higher cooperation levels enhance coalescence success.

correlated, mediated by the structure of the $C$ and $D$ matrices. Fig 2B shows that the difference between the proportion of m-preferences consumers before and after assembly ($\Delta m_r$, Methods; Step 1), increases with $m_r$ for all simulated values of $k_c$, indicating that more generalist species are less prone to extinction during assembly. For the lowest value of $k_c$, $\Delta m_r$ is a monotonically increasing function of the number of preferences. This is expected because a species able to harvest energy from multiple resource pools is less likely to go extinct during community dynamics. As $k_c$ increases, $\Delta m_r$ displays a minimum (Fig 2B), indicating that in more competitive environments pure specialists become more prevalent than moderate generalists. This is because in a highly competitive environment the resource demands are concentrated on a subset of resources, while others are barely consumed (Section 2 in S1 Text). In these communities, consumers that specialize exclusively in empty niches thrive. Fig 2C shows that specialist consumers are systematically present in higher abundance than generalists for all values of $k_c$. This is because several specialists deplete all resources through their combined action more efficiently than one generalist [25], and as a result, although generalists are more persistent than specialists upon assembly (Fig 2B), they achieve lower abundances at equilibrium (Fig 2C). In Fig 2B (inset) we plot $w_{m_r}\Delta m_r$, that is, we weight Eq 5 by the abundance fraction of each m-preferences species group (Eq 6) (Methods; Step 2). This reveals an optimal group of consumers with a number of metabolic preferences that maximizes both survival probability and abundance at equilibrium. This optimal value increases for more competitive environments (as $k_c$ increases). Finally, Fig 2D shows that more competitive communities tend to be less species-rich, as expected from general competition theory.

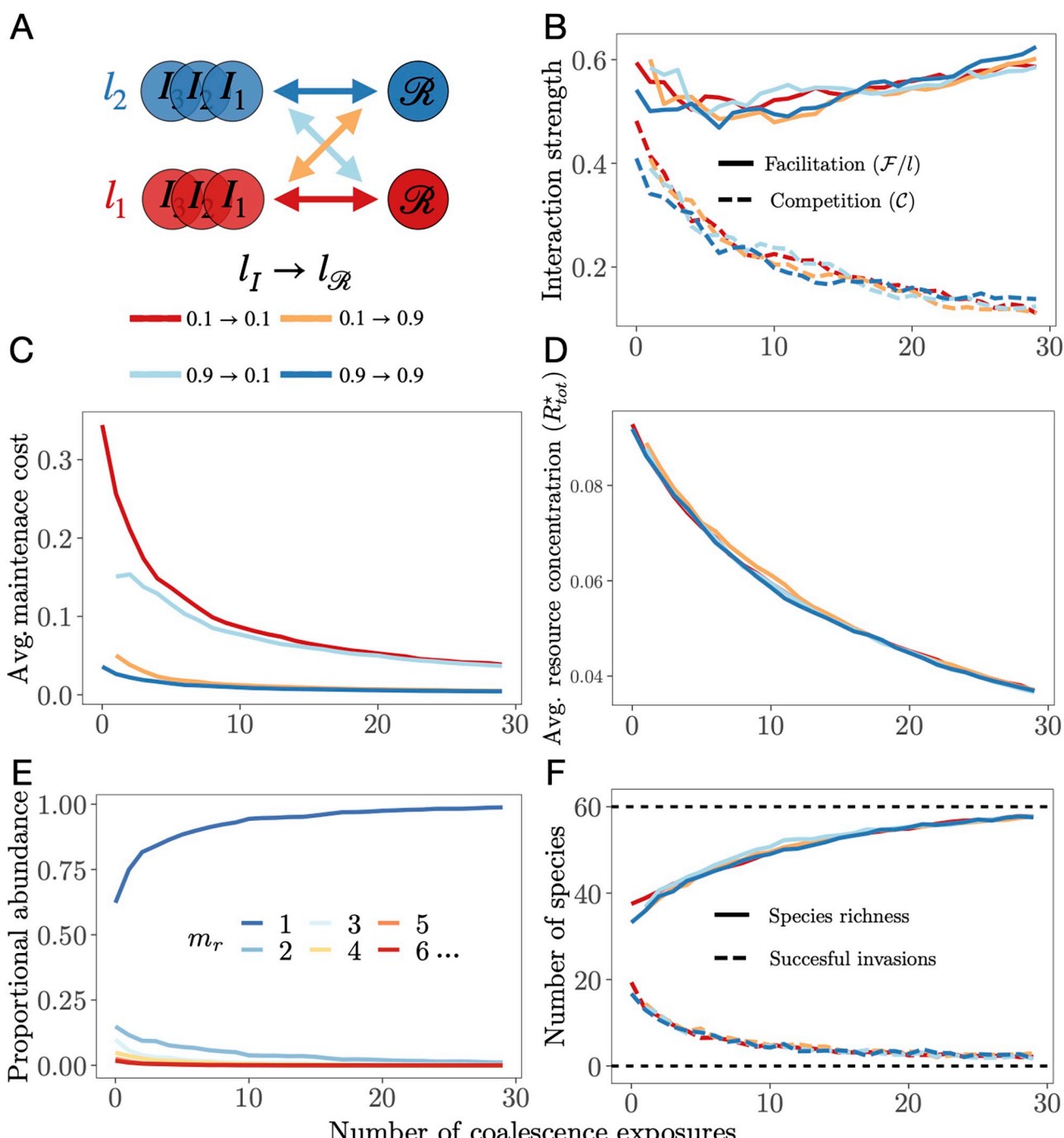

**Fig 5. Serial coalescence of microbial communities. A**: Sketch of the simulation set up. Resident communities ($\mathcal{R}$, upper circles) with leakage $l_{\mathcal{R}}$ are successively coalesced with randomly sampled invader communities ($\mathcal{I}$, lower circles) with leakage $l_{\mathcal{I}}$ for all possible combinations of leakage values (arrows) $l_{\mathcal{I}} = l_{\mathcal{R}} = [0.1, 0.9]$. For each serial coalescence sequence, we examine as a function of number of coalescence events, the following community properties of $\mathcal{R}$: (**B**) community-level competition ($\mathcal{C}$, dashed lines) and facilitation ($\mathcal{F}$, solid lines); (**C**) average species maintenance cost; (**D**) average resource concentration at equilibrium; (**E**) abundance fraction of each m-preferences species group; and (**F**) number of successful invasions and species richness. All the measures are averaged across 20 replicates. The standard deviation is decreasing along the x axis and is never more than 40% of the mean for all the curves (not shown to reduce clutter). Abundance fraction of species with $m_r > 5$ was negligible and is not plotted for clarity.

### Reducing competition increases coalescence success

Fig 3 shows that communities with lower net competition values tend to perform better in coalescence as seen by the positive relationship between parent community dominance ($S_{1,2}$) and net competition difference $(\mathcal{C}_1 - \mathcal{F}_1) - (\mathcal{C}_2 - \mathcal{F}_2)$ (Fig 3C). That is, communities that emerge following coalescence tend to be more similar to the less net competitive parent. This trend holds at higher values of leakage, where cooperation levels are significant (Fig 3B), but with a clear critical point (the yellow line reverses in direction at extreme values of net competition difference). This pattern is driven by the fact that less net competitive parent communities deplete resources more efficiently and achieve a higher species richness (Fig 3D; Section 1 in S1 Text). All these results also qualitatively hold for microbial communities that have consumer guild structure (Section 4 in S1 Text).

### Cooperation further enhances coalescence success

Fig 4 shows that when a community (*B*) whose leakage fraction increases successively during recursive coalescence events with another (*A*) with a fixed leakage level, the former becomes increasingly dominant. The result is consistent for a range of leakage values of community *A*. This shows that increasing cooperation levels enhance coalescence success.

### Community evolution under repeated coalescence events

Fig 5 shows that on average, competition level significantly reduces and facilitation level increases during repeated coalescence events. Along with this, the average maintenance cost of species present in the resident community decreases with the number of coalescence exposures, and so does average resource abundance at equilibrium (Fig 5C and 5D), indicating that resource depletion ability improves in the process. In addition, the sub-population of resource specialists (that consume only one resource) increases with the number of coalescence events, while the rest of the species groups decrease in abundance (Fig 5E). Finally, the number of successful invasions into the resident community decreases with the number of coalescence events, while its species richness increases (Fig 5F). Taken together, these results show that communities composed of non-competing specialists that cooperate among themselves (Fig 5B and 5E) and reduce their respective metabolic costs (Fig 5C), improve their overall resource depletion ability (Fig 5D). This, in turn, makes them more resistant to multi-species invasions and therefore more successful in pairwise coalescence events (Fig 5F).

## Discussion

Our findings offer new mechanistic insights into the dynamics and outcomes of microbial community coalescence by explicitly considering the balance between competition and cooperation; two key interactions of real microbial communities [25, 33]. Specifically, we find that communities harboring less competing and more cooperative species (that is, having lesser net competition) dominate after coalescence because they are better at depleting resources and resisting invasions. Therefore, when a community undergoes a series of coalescence events, its competitiveness decreases and cooperativeness increases, along with its species richness, resource use efficiency, and resistance to invasions. These results provide a theoretical foundation for hypotheses suggested recently [17, 20], and mechanistic insights into empirical studies that have demonstrated the importance of cross-feeding interactions on community coalescence [21].

Our result based on coalescence between pairs of random communities at very low leakage (black line in Fig 3C) essentially extends the results of [24] to communities with both

competitive and cooperative interactions. Tikhonov showed that coalescence success is predicted by minimizing community-level competition through the optimization of resource niche partitioning, which also guarantees maximization of resource depletion efficacy. Here we show that, similarly, the successful community is the one that achieves lower *net* competition ($\mathcal{C} - \mathcal{F}$), which also predicts community-level resource depletion efficacy as well as species richness (Fig 3D). Thus, simultaneously reducing competition and increasing cooperation together drives the outcome of community coalescence. Therefore, the quantity $-(\mathcal{C} - \mathcal{F}) = \mathcal{F} - \mathcal{C}$ is also a measure of the "cohesiveness" of a microbial community. However, we also find that at extreme values of leakage ($l = 0.9$), there is a critical level of net competition difference beyond which coalescence success decreases again (yellow line in Fig 3C). This suggests that in the regime of high cooperation and competition, (high leakage, and tail ends of the curve, respectively) facilitative links become detrimental. A similar result has been reported in [34]. This critical value is not seen when the cost function does not include leakage (Fig B in S1 Text). Interestingly, we also find that this phenomenon is very weak when biologically-realistic guild structure is present (Fig F in S1 Text). These effects of extreme leakage (and facilitation) on coalescence success cannot be predicted by our model analyses (Section 1 in S1 Text), and merit further investigation in future research, provided such high leakage levels are biologically feasible.

In our model systems, species compete not only for resources leaked by other species but also for resources leaked by themselves, i.e., species may leak metabolic by-products that are also encoded in their consumer preferences vector. Leakage of metabolic resources is a pervasive phenomenon in the microbial world [35, 36], and has been shown to exist also in resources necessary for growth, even in situations when those essential metabolites are scarce [37, 38]. Although it may seem counter-intuitive for microbes to secrete metabolites essential for their growth, such leakage can be advantageous, especially in bacteria, as "flux control" or growth-dilution mechanisms, which provide short-term growth benefits in crowded environments [39, 40].

Our recursive coalescence simulations (Fig 4A) allowed us to establish that coalescence success is enhanced by cooperative interactions. This result is consistent with past theoretical work showing that mutualistic interactions are expected to increase structural stability by decreasing effective competition [41]. It is also consistent with recent *in silico* results on single species invasions in microbial communities [42]. Nonetheless, this finding hinges on our choice of the cost function (Eq 2; Section 1 in S1 Text). This cost function, which was motivated by biological considerations, imposed an efficiency cost to species with lower leakage, ensuring that all consumers, independently of their leakage fraction, depleted resources to the same concentration on average (see Section 1 in S1 Text). This allowed us to perform coalescence events between communities harboring species with different leakage without introducing a bias towards the more efficient species. This choice corresponds biologically to the interpretation of leakage as an efficiency factor in the conversion from energy to biomass (Eq S2 in S1 Text). As a consequence, higher leakage species reach, in general, lower abundances at equilibrium [43].

Our findings regarding the evolution of community-level properties in response to repeated community-community encounters (Fig 5) suggest that it might be possible to identify functional groups of microbes or microbial traits that are a "smoking gun" of past coalescence events experienced by a given community [17]. Additionally, our finding that members of communities with a history of coalescence are likely to become increasingly resistant to further community invasions suggests a novel and potentially economical way to assemble robust microbial communities. We also found that repeated coalescence events contributed to

increasing species richness, offering another mechanism that may help explain differences in microbial diversity across locations and environments [17].

Our finding that resident communities exposed to repeated community invasions were mainly composed of cooperative specialists (Fig 5B and 5E) is due to our assumption that all resources were supplied, and at a fixed rate. This allowed specialists to survive because their only source of energy was always provided. This property may not be as commonly seen in real communities, where fluctuations in resource supply are common. Ignoring environmental fluctuations allowed us to focus on coalescence outcomes in terms of the species interaction structure alone. While this assumption may be sensible in some cases [44], it is an oversimplification in others [32]. Therefore, studying the complex interplay between biotic interactions and environmental factors, e.g., by allowing substrate diversification from a single supplied resource [6, 7], or perturbing the resource supply vector periodically to simulate some form of seasonality, is a promising direction for future research. In such cases, we expect a more balanced mix of generalists and specialists, such that only the competitive interactions necessary to diversify the available carbon sources will persist upon coalescence events, but above that threshold, the results presented here (Figs 3, 4 and 5) would be recovered.

Assuming a core leakage and metabolism common to the whole community, made the assembly dynamics computationally tractable while ensuring that the system was not far away from the conditions of real communities [6, 26]. However, the assumption of common leakage was relaxed in the recursive coalescence procedure. In addition, both the community-wide fixed leakage and core metabolism assumptions were relaxed in the serial coalescence simulations, thus using the model in its fully general form as seen in Eq 1. Finally, to retain an analytically tractable theoretical setting for an otherwise complex system, we assumed binary consumer preferences, linear consumer functional responses, and resources of equal energetic value throughout. While this is a promising avenue for future work, we expect our results to be qualitatively robust to relaxation of these constraints, based on recent work on microbial community assembly dynamics using the same general model [6, 26].

Encounters between microbial communities are becoming increasingly frequent [45], and mixing of whole microbial communities is gaining popularity for bio-engineering [46], soil restoration [47], fecal microbiota transplantation [48, 49], and the use of probiotics [50]. We present a framework that relates the structure of species interactions in microbial communities to the outcome of community coalescence events. Although more work is required to bridge the gap between theory and empirical observations, this study constitutes a key step in that direction.

## Supporting information

**S1 Text. Section 1**: Further details of the mathematical model. **Section 2**: Modulating net competition levels. **Section 3**: Further details of the community coalescence simulations. **Section 4**: Adding consumer guild structure. **Fig A**: Consequence of relaxing the assumption of leakage dependence. **Fig B**: Results of random coalescence procedure without imposing the efficiency cost (leakage dependence). **Fig C**: Eigenvalues of J when evaluated at equilibrium for two example communities. **Fig D**: Examples of differently-structured preference (*C*) and metabolic (*D*) matrices. **Fig E**: Auto-correlation of vector of species abundances in community B for consecutive re-assemblies of this community, along the studied leakage range. **Fig F**: Community coalescence with consumer guilds present.
(PDF)

## Acknowledgments

We want to acknowledge Mikhail Tikhonov, Stefano Allesina, members of the Allesina Lab; Zachary Miller, Paulinha Lemos-Costa, and Katja Della Libera, and colleagues; Kiseok Lee, Will Koval, Yolanda Lechón and Maryn Carlson, for their valuable input on early versions of the manuscript.

## Author Contributions

**Conceptualization:** Pablo Lechón-Alonso, Tom Clegg, Jacob Cook, Thomas P. Smith, Samraat Pawar.

**Formal analysis:** Pablo Lechón-Alonso.

**Funding acquisition:** Samraat Pawar.

**Investigation:** Pablo Lechón-Alonso.

**Methodology:** Pablo Lechón-Alonso.

**Software:** Pablo Lechón-Alonso.

**Validation:** Pablo Lechón-Alonso.

**Visualization:** Pablo Lechón-Alonso.

**Writing – original draft:** Pablo Lechón-Alonso.

**Writing – review & editing:** Pablo Lechón-Alonso, Tom Clegg, Jacob Cook, Thomas P. Smith, Samraat Pawar.

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
