## [Decision Letter · Decision Letter 0]

10 Jul 2021

Dear Mr. Lechón-Alonso,

Thank you very much for submitting your manuscript "The role of competition versus cooperation in microbial community coalescence" for consideration at PLOS Computational Biology.

As with all papers reviewed by the journal, your manuscript was reviewed by members of the editorial board and by several independent reviewers. In light of the reviews (below this email), we would like to invite the resubmission of a significantly-revised version that takes into account the reviewers' comments.

As you will see from the reviewers comments, they all found your work interesting and important. However, they also raised some significant questions and concerns which I would hope you will be able to address in a revised version of your manuscript. In your revised manuscript, please provide a detailed point by point response to the reviewers comments.

We cannot make any decision about publication until we have seen the revised manuscript and your response to the reviewers' comments. Your revised manuscript is also likely to be sent to reviewers for further evaluation.

Sincerely,

Daniel Segrè, Ph.D.

Guest Editor

PLOS Computational Biology

Jason Papin

Editor-in-Chief

PLOS Computational Biology

As you will see from the reviewers comments, they all found your work interesting and important. However, they also raised some significant questions and concerns which I would hope you will be able to address in a revised version of your manuscript. In your revised manuscript, please provide a detailed point by point response to the reviewers comments.

Reviewer's Responses to Questions

**Comments to the Authors:**

Reviewer #1: This paper represents an very interesting study about microbial community coalescence in presence of cross-feeding using consumer-resource models. It is very well written and scientifically I have little to add to the paper, so I strongly support its publication. If anything, there is two points that the authors could discuss a bit more in depth and would make the paper in my opinion better.

1) Community cohesion is defined as the difference between facilitation and competition within a community. However, these metrics are computed on the initial pools, i.e. before the community assembly and stabilization steps. Because both competition and facilitation interactions can change during community assembly (e.g. species become extinct, modify their density etc) I am left wondering how the reported results would change if cohesion was measured in the stabilized communities.

2) Similarly, the authors consider equal growth rates for all resources and species (e.g. binary C matrices). I am siilarly left wondering how would relaxing this assumption and considering different growth rates for different species and resources impact the results.

Reviewer #2: Understanding community coalescence, the process through which two communities come together and form a new community, is a current challenge in microbial ecology with important implications in environmental and human health. Microbial communities are often massively dispersed from their original environment (e.g. carried out by macroscopic organisms or weather events), resulting in the sudden encountering of notably different microbial communities. In these cases, we still have little knowledge on how the local community is impacted, in the long term, by the newly arrived community. Similarly, the community traits that are selected after coalescence events are not well understood.

Inspired by recent models (especially, Tikhonov 2016 and Marsland 2019), the authors developed a consumer-resource model that accounts for cross-feeding (or leakage of resources that community members can consume), and tunable ratios of competition-cooperation interactions in multispecies communities. They propose new metrics to assess the cohesion of the resulting in silico communities, and they use these metrics to assess the success of parent communities in the outcome of coalescence events. This success, assessed by community similarity measurements, can be understood as the fraction of species in the parental communities that survive in the coalesced community. For the broader part of parameter region considered in the study, they show that communities exhibiting low competition levels are generally more successful in the event of coalescence. However, in the regime where competition is negligible and leakage is particularly high, the more cooperative community tends to be less successful (or more invadable). The model relies in a number of assumptions that, to me, seem generally reasonable in order to qualitatively address a range of generic scenarios. In my opinion, the work is interesting and sound to a broad audience in microbial ecology and, in my opinion, worth publishing in PLOS Computational Biology after some moderate revision.

I write below a few suggestions that the authors should consider before resubmitting their manuscript:

MAJOR COMMENT:

- The results strongly focus on the success of parental communities at surviving the coalescence process. However, I was missing a comparison between the properties of the parental communities and the coalesced community. The only results shown in that regard are to be found in supplement’s Fig S2. Are coalesced communities more cohesive than both parental strains, or something in between? Do facilitation/competition levels increase or decrease during coalescence? I think that there is an important discussion missing that could shed light on community-level properties that might, or might not, be optimized during coalescence. If we iterate coalescence for a few rounds (the survival species of two coalesced communities are exposed to a new coalescence event), what should we expect? What is the speed at which iterative coalescence leads to a sort of convergence in community structure? I think that adding a figure addressing (at least some of) these questions in the main text could improve the significance of the work.

MINOR COMMENTS:

- The second paragraph in the introduction seems to rush directly into the importance of interactions in coalescence. Around line 14, I think that it could be worth to introduce other potential drivers of coalescence. Just as possible examples to address: resource availability, species growth rates, dispersal rates (communities in the process of coalescence might still receive some degree of immigration from the environment), seasonality (or temperature, day-night light cycles, …).

- Eq. 2: wouldn’t it be better to add subindexes to the noise term epsilon? Otherwise it looks like a universal constant in the equation. In any case, the supplementary text S2 could be a bit more precise about the parameters of the gaussian distribution from which epsilon is sampled, since the text in S2 only says that is a ‘small fluctuation term’. How much small? How sensitive the system is to this noise?

- The meaning of kappa in Eq 5 should be briefly introduced in the main text, even if it’s explained in more detail in the supplement.

- Before Eq. S4: why is it that consuming resource j is relevant to assess competition on resource k? It seems to me that a third species could be consuming j and leak k, and then the two focal species compete just for consuming k.

- I also have a hard time at understanding why competition for leaked resources and supplied resources ‘need to be calculated differently’ (page 4 in the supplement), meaning separately. I understand that it’s convenient to compute the two terms separately. Do the authors mean ‘can be calculated separately’ instead of ‘needs to be…’?

- Why is it that (1-l) affects the metric for abiotic competition? In principle, microbes compete to uptake a full unity of a given resource from the environment. Once this happens, microbial metabolism converts (1-l) into population density, and a fraction ‘l’ is leaked in the form of other resources. But I don’t think that competition itself should be weighted by the unleaked fraction in this case.

- The amount of parameters needed to describe the model makes it a bit hard for the reader to remember the meaning of each of them in the first reading. To make the paper more rapidly understandable at first glimpse, the authors could consider adding legends or equations for each term appearing on the figures. For example a legend in Fig 1 could include ‘cohesion = facilitation – competition’, ‘D = metabolic matrix’. Same for the meaning of ‘kc’, ‘Kc’ and ‘r’ in Fig 2 and 3.

- In figure 3D, R*/r is named ‘Resource depletion level’. Through this name, I would understand that a higher value on this quantity translates to a lower R*. That is, communities with higher resource depletion levels should more efficiently deplete the resources. Unless I am missing something, this is not how it is interpreted and, to me, it is a little bit counterintuitive in the present form.

- I think that fig 4A is not referenced in the main text.

Reviewer #3: The review is uploaded as an attachment

**Have the authors made all data and (if applicable) computational code underlying the findings in their manuscript fully available?**

Reviewer #1: Yes

Reviewer #2: Yes

Reviewer #3: Yes

PLOS authors have the option to publish the peer review history of their article (what does this mean?). If published, this will include your full peer review and any attached files.

Reviewer #1: **Yes: **Djordje Bajic

Reviewer #2: No

Reviewer #3: **Yes: **Alberto Pascual-García
---

## [Decision Letter · Decision Letter 1]

5 Oct 2021

Dear Mr. Lechón-Alonso,

Thank you very much for submitting your manuscript "The role of competition versus cooperation in microbial community coalescence" for consideration at PLOS Computational Biology. As with all papers reviewed by the journal, your manuscript was reviewed by members of the editorial board and by several independent reviewers. The reviewers appreciated the attention to an important topic. Based on the reviews, we are likely to accept this manuscript for publication, providing that you modify the manuscript according to the review recommendations.

As you will see from their comments, the reviewers overall found that your revised manuscript addresses most of their concerns.

One reviewer, however, raised a few additional points that I would ask you to please address in a further revision (including the code availability point) before I can provide my final recommendation for publication.

Sincerely,

Daniel Segrè, Ph.D.

Guest Editor

PLOS Computational Biology

Jason Papin

Editor-in-Chief

PLOS Computational Biology

[LINK]

As you will see from their comments, the reviewers overall found that your revised manuscript addresses most of their concerns.

One reviewer, however, raised a few additional points that I would ask you to please address in a further revision (including the code availability point) before I can provide my final recommendation for publication.

Reviewer's Responses to Questions

**Comments to the Authors:**

Reviewer #1: The authors have adressed my concerns and I can now reccomend the paper for publication.

Reviewer #2: In my opinion, the authors have carefully addressed all the questions and suggestions from the three referees, and I think that the quality of the work has significantly improved during the process. I recommend the current version for publication in PLOS Computational Biology.

Reviewer #3: I would like to thank the authors for the efforts they made to address the questions I raised. I’ve found their results very interesting and I think it is a sound contribution to the field. I would like to ask them to consider the following additional comments, most of which are related to the analysis of parent communities.

Lines 76-83. I do not find convincing the authors’ arguments convincing regarding their choice of the functional form for the secretion of metabolites. In Dwyer’s model, it would be perfectly possible to select appropriate consumption and secretion matrices to account for points i) and ii). For me, the difference is whether secretion of resources is proportional to the growth rate of the species or to their abundances. Since this point is beyond the scope of the article, I would suggest removing this explanation.

Lines 133-135 A brief explanation of the different factors (kc, kf, Kc, Kf) in the Main Text is needed, in particular providing an insight into how they relate to competition / cooperation.

Table 1. Please include in the Table the units of N and R. Also, a comment on the transformation from moles of resource to energy/biomass is needed.

Lines 148-149 I find it difficult to interpret this quantity and the abundances proportion. Firstly, it is not completely clear if are computed considering all the simulations aggregated or if it was computed for each simulation and then averaged (which, with the s.e. I think it would perhaps be more informative). The second thing is that we don’t know which is the distribution of nr, which I think it is needed to understand Fig 2B. For example, it is not clear if it is the case that generalist species are more robust as stated in Results, or if it is just that (only) the more specialized species go extinct and that each species that has more nr preferences jumps to another. (smallest) nr group. The most direct representation I can think of could be a heatmap with nr0 rows and nrFinal in columns (including nr = 0 as extinctions), and the color being the probability that a species in group nr0 ends up in group nrFinal .

Fig 2 Caption:“communities are significantly more cooperative” I would say that this appears to be true for kc = 0.9 (although difficult to say for kf = 0.99), did the authors perform a statistical test to state that it is significant? Moreover, it is not possible to tell anything just from this figure for kc = 0.01.

Fig 2. Please provide the formula used to compute the Inset (perhaps as SM).

Caption Fig. S3. Please indicate the order of magnitude of the leading eigenvalue, since the scale covers several orders of magnitude it would be said that it is zero otherwise.

I appreciate the efforts made in the SM to explain with pseudocode the algorithms, but please push the new code in the repo. Also, I would like to suggest creating a release and permanently storing it in Zenodo for making it citable, see: https://guides.github.com/activities/citable-code/

**Have the authors made all data and (if applicable) computational code underlying the findings in their manuscript fully available?**

Reviewer #1: Yes

Reviewer #2: Yes

Reviewer #3: Yes

PLOS authors have the option to publish the peer review history of their article (what does this mean?). If published, this will include your full peer review and any attached files.

Reviewer #1: No

Reviewer #2: **Yes: **Daniel R. Amor

Reviewer #3: **Yes: **Alberto Pascual-García

Figure Files:

Data Requirements:

Reproducibility:

References:

---

## [Editor Report · Decision Letter 2]

22 Oct 2021

Dear Mr. Lechón-Alonso,

We are pleased to inform you that your manuscript 'The role of competition versus cooperation in microbial community coalescence' has been provisionally accepted for publication in PLOS Computational Biology.

Best regards,

Daniel Segrè, Ph.D.

Guest Editor

PLOS Computational Biology

Jason Papin

Editor-in-Chief

PLOS Computational Biology

Thank you for submitting your further revision of the manuscript, and for documenting your modifications, which address the remaining concerns by one of the reviewers.

---

## [Editor Report · Acceptance letter]

4 Nov 2021

PCOMPBIOL-D-21-00903R2 

The role of competition versus cooperation in microbial community coalescence

Dear Dr Lechón-Alonso,

I am pleased to inform you that your manuscript has been formally accepted for publication in PLOS Computational Biology. Your manuscript is now with our production department and you will be notified of the publication date in due course.

With kind regards,

Zsofia Freund
